# *Porphyromonas gingivalis* Peptidyl Arginine Deiminase (PPAD) in the Context of the Feed-Forward Loop of Inflammation in Periodontitis

**DOI:** 10.3390/ijms241612922

**Published:** 2023-08-18

**Authors:** Zsombor Prucsi, Agnieszka Zimny, Alicja Płonczyńska, Natalia Zubrzycka, Jan Potempa, Maja Sochalska

**Affiliations:** 1Department of Microbiology, Faculty of Biochemistry, Biophysics and Biotechnology, Jagiellonian University, 30-387 Krakow, Poland; 2Doctoral School of Exact and Natural Sciences, Jagiellonian University, 30-387 Krakow, Poland; 3Department of Oral Immunity and Infectious Diseases, University of Louisville School of Dentistry, Louisville, KY 40202, USA

**Keywords:** periodontitis, *Porphyromonas gingivalis*, Bcl-2 family, apoptosis, neutrophils

## Abstract

Periodontitis is a widespread chronic inflammatory disease caused by a changed dysbiotic oral microbiome. Although multiple species and risk factors are associated with periodontitis, *Porphyromonas gingivalis* has been identified as a keystone pathogen. The immune-modulatory function of *P. gingivalis* is well characterized, but the mechanism by which this bacterium secretes peptidyl arginine deiminase (PPAD), a protein/peptide citrullinating enzyme, thus contributing to the infinite feed-forward loop of inflammation, is not fully understood. To determine the functional role of citrullination in periodontitis, neutrophils were stimulated by *P. gingivalis* bearing wild-type PPAD and by a PPAD mutant strain lacking an active enzyme. Flow cytometry showed that PPAD contributed to prolonged neutrophil survival upon bacterial stimulation, accompanied by the secretion of aberrant IL-6 and TNF-α. To further assess the complex mechanism by which citrullination sustains a chronic inflammatory state, the ROS production and phagocytic activity of neutrophils were evaluated. Flow cytometry and colony formation assays showed that PPAD obstructs the resolution of inflammation by promoting neutrophil survival and the release of pro-inflammatory cytokines, while enhancing the resilience of the bacteria to phagocytosis.

## 1. Introduction

The World Health Organization (WHO) has estimated that nearly 10% of the global population is affected by periodontitis, an inflammatory disease progressing in the presence of a complex, dysbiotic oral microbiome [1]. The species of microbes associated with periodontitis vary widely, as do its intrinsic and extrinsic risk factors [2,3]. Because of its prevalence in tissue samples of patients with periodontitis, the Gram-negative *Porphyromonas gingivalis* is considered a key pathogen [4]. 

Rheumatoid arthritis was recently reported to be a comorbidity of periodontitis [5], with many similarities in their pathogenesis and progression of these two conditions [6]. Peptidyl arginine deiminase (PPAD), a distinct virulence factor of *P. gingivalis,* has emerged as a key link between these two diseases. This enzyme, which is unique among prokaryotes, catalyzes the post-translational conversion of arginine to peptidyl citrulline [7,8,9], with the resultant citrullinated proteins subject to destructive immune responses. *P. gingivalis* is associated with a wide range of diseases and pathological conditions [10]. Nevertheless, the impact of PPAD on neutrophils and its contribution to the development of periodontal diseases remain incompletely understood.

PPAD can promote biofilm formation both in single- and dual-species systems enhancing the *P. gingivalis* survival rate under aerobic conditions [11,12]. This enzyme has also been associated with direct immunomodulation. Compared with wild-type strain, a PPAD-deficient strain reduced periodontal inflammation and erosive bone damage in a model of experimentally induced arthritis [13]. Prolongation of neutrophil inflammatory responses has been shown to be essential in the development of chronic inflammation. 

Neutrophils, also known as polymorphonuclear leukocytes (PMNs) are the most abundant immune cell type in blood, circulating in a dormant state. Upon stimulation by a chemoattractant, neutrophils cross the endothelium to enter the infected site [14]. Pathogens are captured and eliminated using multiple strategies (Appendix A), such as the formation of neutrophil extracellular traps (NETs), the internalization of invaders by phagocytosis, and the release of the contents of neutrophil granules (e.g., receptors, proteases, and enzymes) [15,16,17]. 

Apoptosis is a type of programmed cell death that guarantees control and balance of the numbers of neutrophils, triggering the resolution of inflammation and preventing extensive damage to the surrounding tissue [18]. In the absence of contrary stimuli, neutrophils rapidly undergo structural changes induced by pro-apoptotic stimuli, such as cell surface relocation of phosphatidylserine, directing PMN-induced elimination of pathogens and their uptake by macrophages [19,20]. 

During infection, the expression of anti-apoptotic proteins is upregulated, prolonging cell survival and blocking the resolution of inflammation. Although *P. gingivalis* may delay PMN apoptosis, the involvement of specific Bcl-2 family members has not been determined. The pro-survival effect of *P. gingivalis* is thought to be due mainly to the presence of lipopolysaccharide (LPS), a cell wall component of Gram-negative bacteria [21]. LPS is one of the most thoroughly studied *P. gingivalis*-associated virulence factors, inducing the expression of pro-inflammatory cytokines and prolonging neutrophil survival [22,23,24]. 

Infection of human dental follicle stem cells with *P. gingivalis* has been shown to induce the expression of PPAD, which has a direct effect on apoptosis inhibition. These effects, however, were not observed when epithelial cells were invaded by *P. gingivalis*, indicating that PPAD-mediated survival is restricted to particular types of cells [25,26]. These findings suggested that PPAD could also affect the expression of anti-apoptotic proteins associated with the delay of PMN apoptosis [27,28]. Although the factors that trigger the development and progression of periodontal disease are highly complex [29,30], the present study focused on periodontitis, the chronic form of the disease, independent of stage or grade. 

## 2. Results

*P. gingivalis* and the anti-apoptotic properties of its virulence factor towards neutrophils were evaluated by analyzing the time-dependence of programmed cell death induced by this bacterium. Unstimulated neutrophils (UN) were incubated with wild-type *P. gingivalis* (ATCC33277 WT) or a mutant lacking the enzymatic activity of PPAD, resulting from the substitution of the catalytic amino acid Cys^351^ by Ala (PPAD mut), at various multiplicities of infection (MOIs) for 3 h, 24 h, and 48 h, and the neutrophil survival rates were determined.

Following incubation for 3 h (Figure 1a), only slight differences were observed. However, after 24 and 48 h, the numbers of Annexin V-negative cells were significantly higher following incubation with WT *P. gingivalis* than with the PPAD mutant (Figure 1b–d). Therefore, the ability of PPAD to contribute to the anti-apoptotic effect in neutrophils was evaluated. Compared with unstimulated cells, the mutant strain of *P. gingivalis* lacking active PPAD slightly increased PMN survival, whereas the WT strain markedly increased PMN survival (Figure 1b–d). There was no incidence of cytotoxic or lytic cell death (Appendix A). After 48 h, the response to WT *P. gingivalis* was not dose-dependent (Figure 1c), most probably due to saturation of the anti-apoptotic cellular response. By contrast, the response to the PPAD mutant strain remained dose-dependent, indicating that the PPAD virulence factor was essential for lasting anti-apoptotic effects. These results indicated that PPAD plays a role in the delayed programmed cell death of PMNs and that the PPAD mutant strain has a weaker impact on pathogen-mediated PMN survival. Moreover, the addition of polymyxin B (PMXB), a highly effective inhibitor of LPS [31], further diminished the *P. gingivalis*-mediated anti-apoptotic effect. A synergistic effect was observed, as neutrophil survival was blocked and corresponded to the level of untreated control (Figure 1d), indicating the significance of these virulence factors, i.e., LPS and PPAD.

As expected, UNs showed a significant, time-dependent decrease in viability. In contrast, neutrophils infected with WT *P. gingivalis* became resistant to cell death (Figure 2a). To determine the involvement of anti-apoptotic proteins in the survival differences upon infection with WT and mutant strains of *P. gingivalis* (Figure 2), the expression of Bcl-2 family members was evaluated by Western blotting [32]. 

The level of Mcl-1 in untreated neutrophils was very low, showing only slight upregulation upon treatment with either strain of *P. gingivalis,* a finding that may be due to the short half-life of this protein [33]. By contrast, Bcl-xl expression was induced upon bacterial challenge, with only minor differences between *P. gingivalis* WT and PPAD mutant strains (Figure 2b). The induction of A1 expression, however, was significantly lower in neutrophils stimulated with the PPAD mutant than with WT *P. gingivalis* (Figure 2b).

Notably, A1 protein level was upregulated by WT *P. gingivalis* in a dose-dependent manner, with a higher MOI associated with higher levels of protein (Figure 2b, Appendix A). Taken together, these findings revealed that Mcl-1, Bcl-xl, and A1 are associated with resistance to neutrophil cell death, which may inhibit the resolution of inflammation.

High levels of secreted pro-inflammatory cytokines are indicators of the chronic inflammatory state characteristic of periodontitis [34]. TNF-α is a pro-inflammatory cytokine that affects many cell-associated pathways. Cytokine release by neutrophils differed significantly at all time-points following incubation with bacterial strains bearing WT and mutant PPAD (Figure 3a), with the greatest difference between the two strains observed after incubation for 3 h with bacteria at an MOI 1:10 (Figure 3a). Secretion of TNF-α was also dependent on the dose of bacteria (Figure 3c). Cytokine release by neutrophils incubated with *P. gingivalis* bearing WT and mutant PPAD at an MOI of 1:100 showed the most significant differences at 24 h and 48 h (Figure 3a). Thus, based on the high numbers of bacteria, accompanied by high concentrations of virulence factors, the observed effect might be attenuated at the beginning of infection.

IL-6 is another important pro-inflammatory cytokine implicated in the progression of periodontal disease [35,36]. Compared with PMNs infected with WT *P. gingivalis,* PMNs infected with the PPAD mutant strain showed reduced secretion of IL-6 after 3, 24, and 48 h (Figure 3b). In addition, secretion of IL-6 was dependent on MOI, suggesting that infection at a higher MOI promoted a stronger immune response. Neutrophil activation did not reach its highest limit at the highest MOI of the mutant strain tested, with these cells still having the capacity to produce additional cytokine production. Compared with TNF-α, the induction of IL-6 was much weaker, suggesting that the differences between the WT and mutant PPAD cell lines might mask the small differences in concentrations and the statistical significance. Overall, these results suggested that PPAD possesses strong immune-modulatory activity due to its regulation of pro-inflammatory cytokine release.

A major component of the host defense mechanism is the phagocytosis of pathogens by neutrophils [16]. Phagocytosis, however, is dependent on both the neutrophils and bacteria, with the ability of neutrophils to phagocytose many bacterial strains being attenuated [37]. Invasion assays were therefore performed to analyze the possible shielding effect of PPAD. Because phagocytosis can trigger the generation of reactive oxygen species (ROS), also known as an oxidative burst, the effects of PPAD on ROS generation were evaluated [38]. 

The phagocytic activity of neutrophils incubated with the PPAD mutant strain was higher than the activity of neutrophils incubated with WT *P. gingivalis* (Figure 4c). This finding indicated that PPAD plays a crucial role in protecting the pathogen against host elimination, providing a positional advantage for *P. gingivalis.*

Because phagocytosis is the primer trigger of oxidative burst, ROS production by neutrophils was evaluated by flow cytometry. Incubation of neutrophils with WT and mutant *P. gingivalis* at different MOIs for 3 h and 24 h showed that both strains of bacteria triggered ROS production, with no significant between-strain differences after 3 h (Figure 4a). After 24 h, however, neutrophils infected with the mutant strain became hyperresponsive. These results indicated that bacteria secreting the catalytically inactive PPAD (PPAD_C351Ala) were internalized significantly more efficiently by neutrophils than bacteria secreting catalytically active PPAD. Moreover, the incubation of neutrophils with PMXB alone for 24 h increased ROS levels (Figure 4b), further indicating that PMXB has a pro-apoptotic effect on neutrophils [39]. The lack of effect on survival rate and the elevated production of ROS observed after the elimination of PPAD activity and LPS deletion indicated that these virulence factors are crucial, both for *P. gingivalis*-mediated immunomodulatory activity and for inhibition of PMN elimination activity.

## 3. Discussion

Periodontitis is a chronic inflammatory disease manifesting as periodontal damage and caused mainly by a shift in the oral microbiome. Manipulative pathogens benefit from the inflammatory environment due to the elevated amount of nutrients available in the niche. In addition, pathogens have developed multiple virulence factors to further exploit the benefits of hyperinflammation [40].

PPAD is an enzyme that catalyzes protein citrullination and is associated with autoimmune responses, explaining the link between periodontitis and rheumatoid arthritis [10]. The proteome and citrullinome characterized to date indicate that PPAD has a much broader role, as this enzyme has been found to target multiple bacterial proteins, such as gingipain RgpA (arginine-specific gingipain A) [41]. This serves as a basis for TLR2-mediated cellular responses to the presence of bacteria [42]. 

Successful microbial invaders are characterized by two main characteristics. The first is their ability to delay neutrophil apoptosis, thereby delaying the resolution of inflammation, as well as having immunomodulatory activity, thereby fueling a pro-inflammatory state. The second characteristic is the ability of these microbes to escape the host’s defense mechanisms. The *P. gingivalis* virulence factor PPAD was therefore tested according to these criteria to assess the role of PPAD in the pathogenesis of periodontal disease. 

Neutrophils are short-lived leukocytes, with the present study showing that, in the absence of stimuli after 24 or 48 h, HoxB8 neutrophils underwent apoptosis. WT *P. gingivalis* prolongs the survival of human neutrophils, an activity mainly attributed to LPS [21]. The present study showed that PPAD directly prolongs the survival of PMNs. Neutrophil survival was significantly shorter in the presence of the LPS inhibitor PXMB than under unstimulated conditions. These results provided crucial baselines for non-autoimmune responses. 

Apoptosis is regulated by multiple pro- and anti-apoptotic proteins, as well as cytokines and intrinsic and extrinsic signals [43]. The present study investigated the contribution of three members of the Bcl-2 family, Mcl-1, Bcl-xl, and A1, which are relevant to neutrophil biology, to apoptosis [44]. Challenge with WT *P. gingivalis* resulted in the significant induction of apoptosis, with A1 expression differing in neutrophils stimulated with WT and PPAD-deficient strains of *P. gingivalis*. These results indicate that the promotion of neutrophil survival is mainly dependent on A1 expression levels. Because these proteins are coordinately regulated at multiple levels, the molecular mechanisms remain to be determined [45]. Although it would be of great interest to assess the impact of a selective and potent A1 inhibitor, this type of inhibitor has not yet been developed (unpublished). Several inhibitors of Bcl-2-like proteins are commercially available, but none of these specifically targets A1/Bfl-1, probably due to the high level of similarity of protein structures similarity. Studies are underway to identify potential targets and antagonists, not only in periodontitis but also in other conditions [46,47].

The chronic inflammatory state characteristic of periodontitis is the result of complex cellular and molecular activities. TNF-α plays a critical role in the control of neutrophil survival [48], both enhancing apoptosis and increasing the expression of Bcl-2-like anti-apoptotic proteins, such as A1/Bfl-1, Bcl-xl and Mcl-1 [49,50,51]. Nuclear factor-kappa B (NF-κB) is a transcription factor involved in the regulation of proteins belonging to the Bcl-2 family and a key regulator of the expression of proteins in the immune system that can be induced by TNF cytokines [52,53]. Upon infection, TNF-α levels are increased both in serum and saliva, indicating an active state of immune responsiveness [54]. 

In addition to TNF-α, several other cytokines, such as IL-6, have been detected at diseased periodontal sites [55,56]. IL-6 is a key cytokine in chronic inflammation [57], due to its ability to delay apoptosis and activate various inflammatory pathways, such as the mitogen-activated protein kinase cascade [58,59]. The effects of IL-6 on neutrophil apoptosis are not unequivocal, as strong concentration dependency has been reported [60]. Both TNF-α and IL-6 are well-established pro-inflammatory cytokines, suggesting that they are essential targets to reveal the molecular pathways affected by PPAD [61] (Figure 5). 

The regulatory activities of TNF-α and IL-6 are highly heterogeneous, complicating investigations of their activities. Their involvement in pro-inflammatory and pro-apoptotic pathways can determine cell fate, as has been indicated schematically. By contrast, under certain not yet fully characterized conditions, pro-survival activity dominates via the upregulation of expression of anti-apoptotic proteins. Delayed apoptosis can lead to uncontrolled neutrophil persistence and delayed resolution of inflammation. The expression of members of the Bcl-2 protein family, including Bcl-xl, A1, and Mcl-1, all of which have a short half-life (20–30 min), is induced by different transcription factors, including NF-κB and STAT [33]. NF-κB, which can be activated by intrinsic or extrinsic pathways (such as LPS), is considered a key transcription factor during inflammation.

The release of pro-inflammatory cytokines during infection indicates activation of the host’s innate immune system [61]. TNF-α release was significantly lower upon infection of neutrophils with PPAD-deficient than with WT *P. gingivalis*. Similarly, IL-6 secretion differed in neutrophils infected with WT and PPAD-deficient *P. gingivalis*. These findings indicated that PPAD strongly contributes to the immunomodulatory activity of *P. gingivalis* through these pro-inflammatory cytokines. Stimulation of neutrophils with WT bacteria induced the production of TNF-α in a time-dependent manner. By contrast, stimulation with the PPAD mutant strain induced a substantial increase in TNF-α production between 3 and 24 h. The diminished neutrophil response might be due to the lower pathogenic activity of the PPAD mutant strain. Unlike TNF-α, IL-6 concentration markedly increased over time in the presence of both bacteria strains. This finding suggests that, at the beginning of the infection when the pathogen is present in low numbers, the observed response might be attenuated to provide sufficient time for colonization.

Prolongation of inflammation and bleeding is crucial for *P. gingivalis*, as it ensures access to an elevated amount of nutrients (e.g., heme from erythrocytes). *P. gingivalis* also requires evading the antibacterial activities of the host immune system. The citrullination of host proteins, such as antimicrobial peptides, signaling molecules, and histones, diminishes host defenses by the innate immune system. This may manifest as a reduced efficiency in capturing pathogens by NETs, providing further evidence for the essential role of PPAD during pathogenesis [62,63]. Additional studies are required to further characterize the shielding effect of PPAD against phagocytosis. In addition to oxygen-independent neutrophil defense mechanisms, the oxygen-dependent effects of PPAD were characterized by measuring ROS generation. WT *P. gingivalis* were more protected from phagocytosis than the PPAD-deficient strain, indicating that PPAD was associated with this protective activity, findings consistent with previous results [62]. Moreover, compared with WT-*Pg*, challenge with the PPAD mutant strain increased the phagocytosis-induced apoptosis of neutrophils, which could explain the diminished expression of the anti-apoptotic proteins A1 and Bcl-xL. The shielding effect of PPAD was further indicated by the significant release of ROS following stimulation with the PPAD-deficient strain, as phagocytosis is the main signal for the release of enzymes catalyzing the production of ROS. Because PPAD activity is required for maximum gingipains activity, the present results are in agreement with results showing decreased phagocytic activity in the presence of RgpA [64].

The long-term goal of this research is to identify and develop novel diagnostic markers for periodontal disease and other chronic inflammatory diseases. This might result in the establishment of new diagnostic tests. At present, the diagnosis of periodontitis is based on clinical examinations, which indicate the degree of tissue damage. These examinations, however, are time-consuming, painful, and subjective [65,66]. Surgical and non-surgical strategies are currently available for the treatment of patients with periodontitis [67]. Non-surgical strategies involve very painful deep cleaning (scaling and root planning), supplemented with medications such as antibiotics [68]. The success rates of these mechanical treatments are limited by the stage of disease and may be followed by the reformation of plaque [69]. Currently available medications alone are somewhat insufficient and cannot replace surgery [70]. New biomarkers are needed for the diagnosis of periodontitis, and more efficient drugs are needed for its treatment.

## 4. Materials and Methods

### 4.1. Cell Lines and Bacteria Strains

Throughout the study, the HoxB8 model system was used [24] (Graphical Abstract (GA)). Mouse-derived neutrophil progenitor cell lines were grown in OPTI-MEM (Gibco) supplemented with 10% FBS, 100 U/mL penicillin and 100 µg/mL streptomycin, 250 µM L-glutamine, 30 mM β-mercaptoethanol, 1 µM β-estradiol (all from Sigma-Aldrich, St. Louis, MO, USA), and Stem Cell Factor (SCF, collected from genetically engineered CHO cell line, [71]). Before stimulation, progenitor cells were seeded without β-estradiol in OPTI-MEM supplemented with 2% FBS, 250 µM L-glutamine, 30 mM β-mercaptoethanol and 2% SCF-containing supernatant for 4 days. 

*P. gingivalis* (ATCC WT and C351A PPAD mut) was grown for 7 days on Brain–Heart Infusion (BHI, Becton Dickinson, Franklin Lakes, NJ, USA) agar plates supplemented with yeast extract, 0.5 mg/mL L-cysteine (reducing agent), 10 µg/mL hemin, 0.5 µg/mL vitamin K and with 1 µg/mL tetracycline (Merck-Millipore, Burlington, MA, USA) for the mutant strain. On day -1, the optical density (OD600) was set to 1.0 in liquid BHI media using a DeNovix (DS-11+) spectrophotometer. The C351A mutant strain (“PPAD mut”) expresses an inactive PPAD enzyme constructed by a mutation in the catalytic site (cysteine to alanine) [72]. 

On the day of the experiment (day 0), differentiated HoxB8 neutrophils were challenged with *P. gingivalis.* Following the selected time point, supernatants were collected for ELISA assay. Cells were collected for viability, ROS production analysis, and Bradford protein quantification using accutase treatment. 

### 4.2. Flow Cytometry

The percentage of viable cells was measured by staining with Annexin-V-APC at 1:400 (BioLegend, #640930, San Diego, CA, USA) or Annexin-V eFluor450 diluted at 1:200 (Invitrogen, #88-8007-74, Waltham, MA, USA) in 1× Annexin Binding Buffer (Invitrogen, #88-8007-74). Alternatively, to determine the amount of generated reactive oxygen species, cells were stained for 20 min at 37 °C with 20 μM 2′,7′-dichlorofluorescein diacetate (DCFH-DA #D6883) (Sigma-Aldrich). The fluorescent signal from DCF^+^ was measured using FACS: Calibur (Becton Dickinson, BD) or LSR Fortessa (BD). Data were analyzed using FlowJo v10 (TreeStar, Ashland, OR, USA) software. 

### 4.3. Immunoblotting

After indicated stimulations, cells were harvested and lysed in CHAPS buffer (as described before, [44]). Protein content was then quantified by Bradford assay (BioShop, Burlington, ON, Canada) and 15–25 µg of protein were loaded on and separated by sodium dodecyl sulfate–polyacrylamide gel electrophoresis (SDS-PAGE) using 12% separating and 4% stacking gel. Proteins were transferred to Immobilon^®^-PSQ membrane (Merck-Millipore) using electro-transfer. The membrane was blocked for one hour in 5% BSA-PBS-Tween 20 or 5% Milk-PBS-Tween 20 (BioShop). The protein of interest was detected using primary antibody against Mcl-1 (Rockland, Pottstown, PA, USA 600-401-394), Bcl-xL (Cell Signaling Technology, #2764, Danvers, MA, USA), A1 (kindly provided by Prof. Marco Herold, WEHI Institute, Melbourne, Australia), GAPDH (Cell Signaling Technology, #2118) and horseradish peroxidase (HRP) conjugated anti-rat/anti-rabbit secondary antibody (Cell Signaling Technology, #7077/#7074). Finally, the membrane was treated with Pierce™ ECL Western Blotting Substrate (Thermo Scientific™, Waltham, MA, USA) and imaged on Hyperfilm (GE Healthcare, Chicago, IL, USA). Pictureswere captured by the ChemiDoc™ MP imaging system (Bio-Rad, Hercules, CA, USA). WEHI-231 cell line was kindly provided by Prof. Andreas Villunger (Medical University of Innsbruck, Innsbruck, Austria).

### 4.4. Enzyme-Linked Immunosorbent Assay (ELISA)

Secreted TNF-α and IL-6 levels were evaluated using standard ELISA carried out with commercially available kits (Mouse IL-6 DuoSet ELISA kit, DuoSet ELISA Ancillary Reagent Kit 2, and Mouse TNF-alpha DuoSet ELISA) according to the manufacturer’s instructions. Absorbance was measured using Flex Station 3 (Molecular Devices, San Jose, CA, USA) at 450 nm and corrected at 570 nm. 

### 4.5. Invasion Assay

The phagocytic activity of neutrophils was assessed by invasion assay. Neutrophils were seeded along with bacteria in MOI:50 for 1 h at 37 °C. Cells were washed twice in PBS and then lysed with distilled water for 20 min. The lysate containing invading bacteria was transferred to a BHI agar plate (without antibiotics) and incubated for 3 days in anaerobic conditions at 37 °C. After 3 days, colonies were counted using an OPTA-TECH^®^ SK diagnostic microscope with a 4.5× magnification. 

### 4.6. Statistical Analysis

Statistical significance (*p* < 0.05) was assessed using an unpaired *t*-test. Analysis was carried out using GraphPad Prism 8.0.1.

## 5. Conclusions

The present study was designed to better understand the contribution of PPAD to the infinite loop of inflammation. Study results showed that the expression of A1/Bfl-1, a neglected Bcl-2 family member, is strongly dependent on PPAD expression by *P. gingivalis*. Inhibition of neutrophil apoptosis hinders the resolution of inflammation, prolonging the chronic state of periodontitis and eroding tooth-supporting tissues. In addition to elevating the expression of anti-apoptotic proteins, PPAD significantly increased the release of pro-inflammatory cytokines, such as TNF-α and IL-6. Moreover, PPAD was found to play a role in shielding *P. gingivalis* from phagocytosis while quenching the release of ROS by neutrophils.

## Figures and Tables

**Figure 1 ijms-24-12922-f001:**
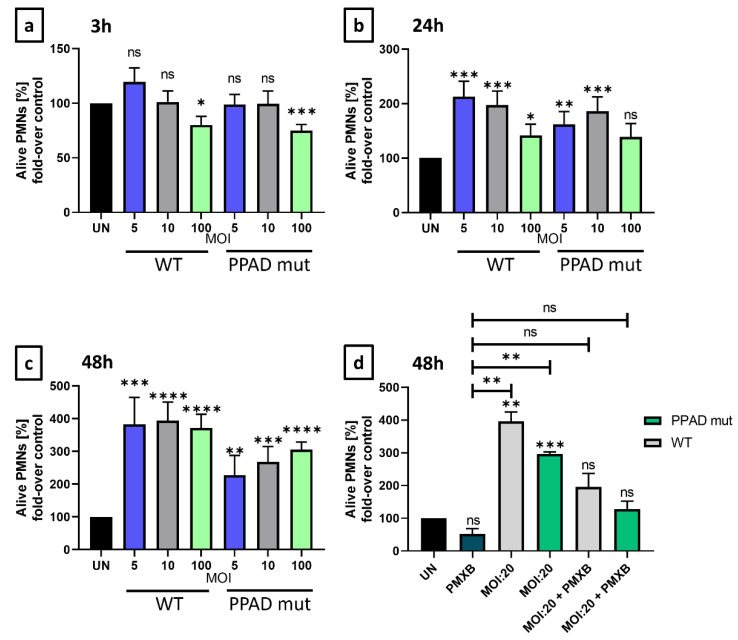
*P. gingivalis* virulence factors, PPAD and LPS, are responsible for anti-apoptotic effects induced upon bacterial challenge. Neutrophil survival was analyzed by flow cytometry and presented as bar plots showing control of Annexin V-negative neutrophils, relative to untreated cells (100% viability). Neutrophils were challenged by either wild-type (WT) or PPAD-deficient (PPAD mut) *P. gingivalis* strains at various MOIs or were left untreated (UN). Neutrophil survival was assessed at (**a**) 3 h, (**b**) 24 h, and (**c**) 48 h after Annexin V staining and fluorescent signal analysis by FlowJo v10 software. Each color represents a different MOI. (**d**) Where indicated, neutrophils were incubated with bacteria and an LPS inhibitor, Polymyxin B (PMXB, 100 µg/mL). Data are presented as means +/− SEM of five sets of duplicate samples * *p* ≤ 0.05, ** *p* ≤ 0.01, *** *p* ≤ 0.001, **** *p* ≤ 0.0001 compared with untreated samples (UN) by unpaired *t*-tests. ns, not significant.

**Figure 2 ijms-24-12922-f002:**
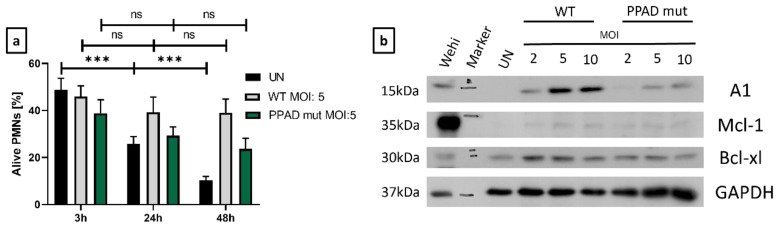
Increased levels of pro-survival proteins are responsible for the anti-apoptotic effects triggered by *P. gingivalis.* (**a**) Flow cytometry analysis of the time-dependence of modulation of neutrophil survival, with bar plots showing Annexin V-negative cells. Neutrophils were incubated with wild-type (WT) or PPAD-deficient (PPAD mut) strains of *P. gingivalis* or left untreated (UN). Results at the indicated time-points and MOIs were evaluated after Annexin V staining and fluorescent signal analysis by FlowJo v10 software. Data are presented as the mean + SEM of five sets of duplicate samples *** *p* ≤ 0.001 by unpaired *t*-tests. ns, not significant. (**b**) Neutrophils were incubated for 3 h with wild-type (WT) or PPAD-deficient (PPAD mut) strains of *P. gingivalis* and lysed in CHAPS buffer, and the expression of anti-apoptotic proteins was assessed by Western blotting. Results are representative of three independent experiments. WEHI-231 (Wehi) cell lysate was used as a positive control.

**Figure 3 ijms-24-12922-f003:**
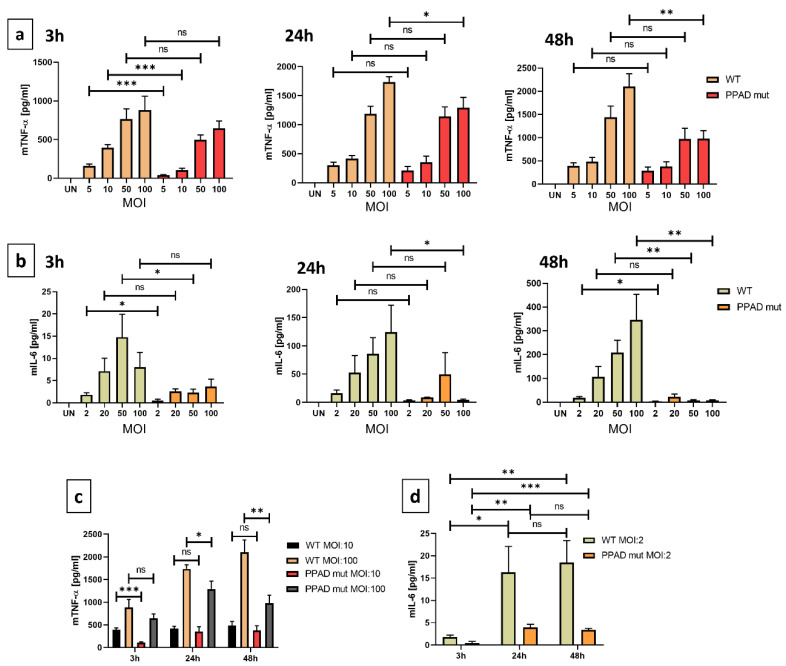
PPAD strongly contributes to the inflammatory responses of infected neutrophils. Supernatants were collected from neutrophils incubated with bacterial strains, and the concentrations of TNF-α (**a**) and IL-6 (**b**) in the supernatants were analyzed by ELISA. Dependence of TNF-α (**c**) and IL-6 (**d**) secretion on MOI and time. Bar graphs show mean +/− SEM cytokine concentrations in five duplicate samples corrected for unstimulated neutrophils (UN = 0). * *p* ≤ 0.05, ** *p* ≤ 0.01, *** *p* ≤ 0.001 by unpaired *t*-tests. ns, not significant.

**Figure 4 ijms-24-12922-f004:**
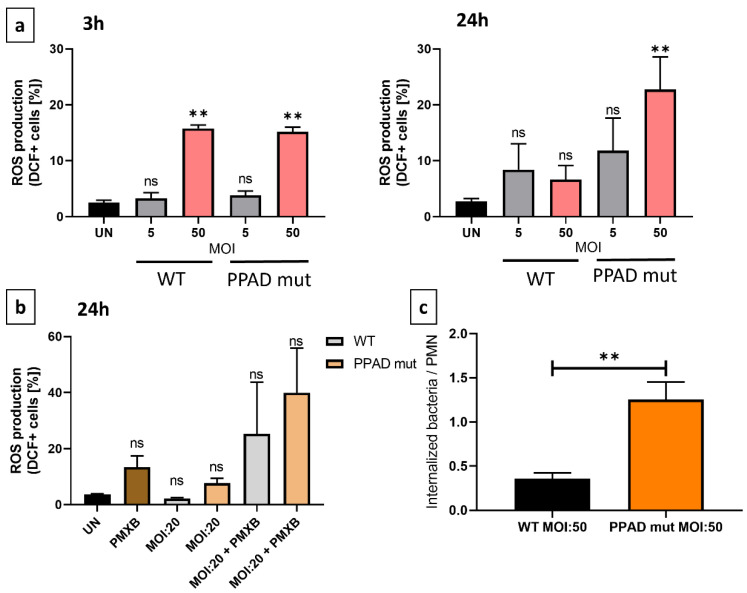
Effects of *P. gingivalis* WT and PPAD mutant on the generation of an oxidative burst in neutrophils. (**a**,**b**) Neutrophils were incubated with wild-type (WT) or PPAD-deficient (PPAD mut) strains of *P. gingivalis* or left untreated (UN), and the release of reactive oxygen species (ROS) was analyzed by flow cytometry. Results at different time-points and MOIs were evaluated after DCFH-DA staining and evaluation of fluorescent DCF^+^ signals with FlowJo v10 software. Where indicated, an LPS inhibitor, Polymyxin B (PMXB, 100 µg/mL), was also administered. Each color represents a different MOI. (**c**) Phagocytic activity and susceptibility, as determined by colony formation-based invasion assays. The bar graphs show the means +/− SEM of five independent experiments. ** *p* ≤ 0.01 byunpaired *t*-tests. ns, not significant.

**Figure 5 ijms-24-12922-f005:**
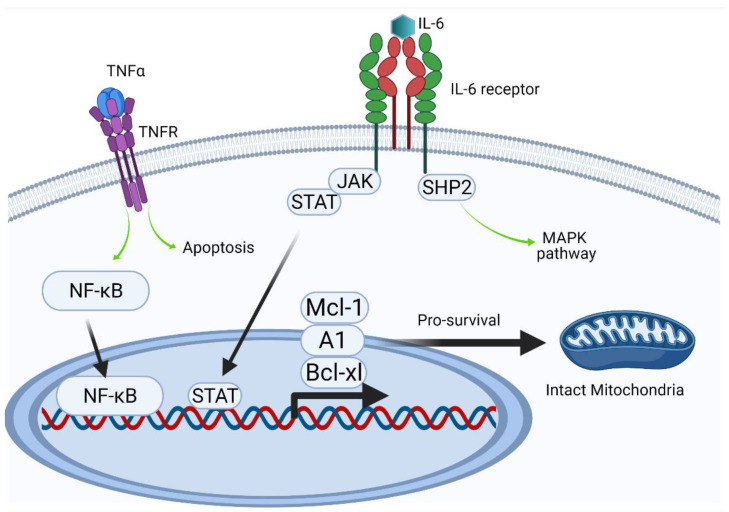
Intracellular effects (Created with BioRender.com).

## Data Availability

Data sharing not applicable to this article.

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
