# Peer review of "Porphyromonas gingivalis Peptidyl Arginine Deiminase (PPAD) in the Context of the Feed-Forward Loop of Inflammation in Periodontitis"

_ijms, 2023, doi:10.3390/ijms241612922_

Round 1
Reviewer 1 Report
The manuscript was well written and has some merits. However, I still have the following major concerns:
1. Figure 1. Perhaps, it will be more beneficial to omit and be replaced at the end to include proposed mechanisms presented with figure 6.
2. Figure 2. Results at 3 and 24 h appeared to be modest. Authors indicated in the text that 24 and 48 h showed significant outcome. However, it was clear to me that at 48 h, the mutant had influenced effect and was dose dependent and the wild type was not dose dependent?. The text did not emphasize this fact. Furthermore, it was stated that "In fact, we could observe a synergistic effect, as neutrophil survival was blocked and corresponded to the level of untreated control (Figure 2d), which proves the significance of these virulence factors. Which factors (LPS or PPAD or both).
* Additional experiment; LDH activity, to verify flow cytometry data.
3. Figure 3. Western blot: Can specify the time point? 48 h? Also, quantification of signals obtained would be more beneficial. " expression of A1 and Bcl-xL was upregulated in a dose-dependent manner by wild-type", yet, dose 10 Bcl disagreed with the this statement?
* Additional experiment to reverse expression or RT-PCR should be included
4. Figure 4. Data was presented well, yet, it appeared to be inconsistent with Fig 2 at 48 h with dose 100? it was shown that more alive cells and more inflammatory cytokines?
5. The conclusion: Results presented did not support statements stated. TNF is pro and IL-6 can be pro and anti inflammatory. Most of stated conclusions were more hypothesis based; "the resolution of inflammation is hindered and the chronic state as well as pathological condition are prolonged" which of their results support this?
Editing service must be used.
Reviewer 2 Report
Review of Porphyromonas gingivalis peptidyl arginine deiminase (PPAD) in the context of the infinite loop of inflammation in periodontitis
This was easy to read and very interesting research where the contribution of peptidyl arginine deiminase to inflammation was examined by comparing inflammatory cell responses induced by wild-type P. gingivalis and PPAD mutant in various assays.
On the whole, the research was well described and interpreted, though in a couple of places the results have been over-interpreted and would require further experimentation to support the interpretation, or requires modification of the manuscript:
Figure 3 and Figure 6. Based on the increased amount of A1 and Bcl-xl protein as the MOI increases, it is assumed that protein expression is increased, but these proteins have short half-lives of 20-30 min. How do you know that there has been increased protein expression rather than a modification of the proteins or environment by P. gingivalis that extends the half-life of the proteins? Showing increased A1 and Bcl-xl mRNA transcripts would have supported an increase in protein expression.
“Overall, our experiments revealed that Mcl-1, Bcl-xl, and A1 are directly associated with resistance to cell death of neutrophils, which most probably inhibits the resolution of the inflammation.” This is inaccurate as you have not shown that they are directly associated with resistance to cell death of neutrophils, instead you have shown an increased amount of these proteins and assumed they are responsible.
Figure 5. The title for Figure 5 is “P. gingivalis PPAD mutant is more susceptible to oxidative burst and internalization by neutrophils”. I don’t agree with PPAD mutant being “more susceptible to oxidative burst”, perhaps the PPAD mutant invoked more oxidative burst?
“Importantly, the PPAD mutant strain was more likely to be internalized and neutralized by neutrophils when compared to the wild-type P. gingivalis strain, as presented in Figure 5c.” What does being neutralized mean, and how do you know? Figure 5c shows the number of internalized bacteria/neutrophil and was determined by plate counting the bacteria once cells were lysed, indicating that the bacteria survived inside the neutrophil.
“Consequently, these results prove that bacteria with inactive peptidyl arginine deiminase were more effectively subjected to elimination by internalization.” Figure 5c shows that the bacteria were internalized but not eliminated as they were detected by viable count.
Regarding the methodology for the Invasion Assay, cells were washed with PBS then lysed. It is possible that P. gingivalis cells could have stuck to the neutrophils and were not washed off, possibly inflating the number of internalized bacteria. Could an antibiotic treatment step have been used to kill all P. gingivalis exterior to the neutrophils prior to neutrophil lysis? Also, for counting the colonies, please detail what model of diagnostic microscope was used and what magnification was required to see the 3 day old colonies, which would not have been visible to the naked eye.
Other comments:
“A distinct virulence factor of P. gingivalis, a peptidyl arginine deiminase (PAD), emerges as a key link. This enzyme catalyzes a post-translational modification, the conversion of protein arginine to protein citrulline, and is found to be unique among prokaryotes” The correct abbreviation given should be PPAD. Also, the second sentence is misleading, as other bacteria can convert protein arginine to protein citrulline using arginine deiminase; PPAD is the only bacterial peptidyl arginine deiminase so converts peptide arginine to peptide citrulline. So it should be “unnatural” citrullinated peptides rather than “unnatural” citrullinated proteins as described in the following sentence.
“PPAD can promote biofilm formation both in single- and dual-species systems enhancing bacterial survival rate under aerobic conditions”. This sentence implies that the dual species system was P. gingivalis and another bacterial species but was actually a fungal species (Candida albicans), so different Kingdom! Use “P. gingivalis” instead of “bacterial” in the sentence.
Gram should always be capitalized when mentioning Gram stains, as they are named after their inventor Hans Christian Gram.
“In fact, we could observe a synergistic effect, as neutrophil survival was blocked and corresponded to the level of untreated control (Figure 2d), which proves the significance of these virulence factors.” It is debatable whether the results showed synergism (combined effect greater than the sum of individual effects).
Where was WEHI-231 purchased from?
“Notably, expression of A1 and Bcl-xL was upregulated in a dose-dependent manner by wild-type P. gingivalis.” Although there is an obvious increase in A1 protein between dose 2 and 5, there does not appear to be any further dose dependent effect between 5 and 10, or for Bcl-xl at all, so don’t generalize about dose dependency.
“Deceptive pathogens benefit from the inflammatory environment due to the elevated amount of nutrients in the niche. To exploit the benefits of a hyperinflammatory state, pathogens developed multiple virulence factors.” “Deceptive pathogens” is a strange choice of words, and all bacteria present would benefit from elevated nutrients. The second sentence sounds like it came straight out of a student thesis; it requires referencing.
Why was cysteine added to agar plates, and what concentration was Tetracycline used at (currently written as 1 ug/M)?
Abbreviation given for 2′, 7′-dichlorofluorescein diacetate of DCFH in the Flow Cytometry section of M&M is different to the abbreviation DCF used in Figures/Results.
Round 2
Reviewer 1 Report
Thank you for addressing my concerns.
The final version shall be edited and reviewed by professional service.
Author Response
Thank you for your suggestions. Once we get the editited manusxeipt from BioEdi, we will upoladed the corrected version (the latest on August 14th).
Reviewer 2 Report
No issues with the manuscript.
Author Response
Thank you.